# Melatonin in Medicinal and Food Plants: Occurrence, Bioavailability, and Health Potential for Humans

**DOI:** 10.3390/cells8070681

**Published:** 2019-07-05

**Authors:** Bahare Salehi, Farukh Sharopov, Patrick Valere Tsouh Fokou, Agnieszka Kobylinska, Lilian de Jonge, Kathryn Tadio, Javad Sharifi-Rad, Malgorzata M. Posmyk, Miquel Martorell, Natália Martins, Marcello Iriti

**Affiliations:** 1Student Research Committee, School of Medicine, Bam University of Medical Sciences, Bam 44340847, Iran; 2Department of Pharmaceutical Technology, Avicenna Tajik State Medical University, 73400 Dushanbe, Tajikistan; 3Department of Biochemistry, Faculty of Science, University of Yaoundé 1, Yaoundé Po. Box 812, Cameroon; 4Laboratory of Plant Ecophysiology, Faculty of Biology and Environmental Protection, University of Lodz, 90-237 Lodz, Poland; 5Department of Nutrition and Food Studies, George Mason University, Fairfax, VA 22030, USA; 6Zabol Medicinal Plants Research Center, Zabol University of Medical Sciences, Zabol 61615-585, Iran; 7Department of Nutrition and Dietetics, Faculty of Pharmacy, University of Concepcion, Concepcion 4070386, Chile; 8Faculty of Medicine, University of Porto, 4200-319 Porto, Portugal; 9Institute for Research and Innovation in Health (i3S), University of Porto, 4200-135 Porto, Portugal; 10Department of Agricultural and Environmental Sciences, Milan State University, 20133 Milan, Italy

**Keywords:** melatonin, bioactive phytochemicals, antioxidants, herbal remedies, tryptophan derivatives, nutraceuticals

## Abstract

Melatonin is a widespread molecule among living organisms involved in multiple biological, hormonal, and physiological processes at cellular, tissue, and organic levels. It is well-known for its ability to cross the blood–brain barrier, and renowned antioxidant effects, acting as a free radical scavenger, up-regulating antioxidant enzymes, reducing mitochondrial electron leakage, and interfering with proinflammatory signaling pathways. Detected in various medicinal and food plants, its concentration is widely variable. Plant generative organs (e.g., flowers, fruits), and especially seeds, have been proposed as having the highest melatonin concentrations, markedly higher than those found in vertebrate tissues. In addition, seeds are also rich in other substances (lipids, sugars, and proteins), constituting the energetic reserve for a potentially growing seedling and beneficial for the human diet. Thus, given that dietary melatonin is absorbed in the gastrointestinal tract and transported into the bloodstream, the ingestion of medicinal and plant foods by mammals as a source of melatonin may be conceived as a key step in serum melatonin modulation and, consequently, health promotion.

## 1. Introduction

Melatonin is a tryptophan-derived substituted indoleamine (*N*-acetyl-5-methoxytryptamine; Figure 1), widely found in living, evolutionarily distant organisms [1]. This molecule was isolated in 1958 by Lerner et al. [2] from cow pineal gland. For decades, melatonin was considered an animal neurohormone with an impact in circadian rhythm regulation, seasonal reproductive cycles, and mammalian immune system modulation [3,4], but currently these studies are outdated. However, in the 1980s and later, melatonin was also identified in many non-vertebrates (e.g., cnidarians, planarians, mollusks, insects, crustaceans) [5,6], and in 1989, it was detected in a phototrophic unicell, the dinoflagellate *Gonyaulax polyedra* (current name: *Lingulodinium polyedrum*) [7], and thereafter analyzed in detail [8]. Subsequently, in 1995, melatonin was identified in some plant species [9,10] and, since then, the interest on this molecule by plant physiologists has rapidly increased. In the 1990s, an essential and novel function was discovered and attributed to this molecule—a highly potent hydroxyl radicals scavenger [11], a finding that was considered the starting point of extensive studies on melatonin antioxidant protection, which turned out to involve numerous mechanisms beyond radical scavenging [12,13,14,15,16,17].

## 2. Melatonin in Plants

Nowadays, it is widely recognized that melatonin is a universal amphiphilic antioxidant molecule, able to penetrate all compartments of a cell because of its small size, and good solubility in both water and lipids. Looking at melatonin antioxidant properties, it mainly consists of (1) direct scavenging of reactive oxygen (ROS) and reactive nitrogen (RNS) species [18]; (2) acceleration of antioxidant enzymes activity [19]; (3) protection against oxidative damage [20]; (4) synergistic effects with other antioxidants [21]; and (5) improvement of the electron transport efficiency in mitochondrial respiratory chain, limiting free radicals overproduction through electron leakage reduction [22,23]. Moreover, not only melatonin, but also its derivatives act as natural electron donors, highly efficient against oxidative stress. Melatonin is also able to generate a radical scavenger cascade (Figure 2), originating oxidation products, such as β-hydroxymelatonin, cyclic β-hydroxymelatonin or cyclic melatonin [24], and *N1*-acetyl-*N2*-formyl-5-methoxykynuramine (AFMK) [25], and to contribute to further ROS elimination [26]. In contrast with other antioxidants, this ability makes melatonin effective in protecting organisms against oxidative stress, even in small doses [27]. Indeed, the maintenance of a proper redox status in aerobic cells (in mitochondria) is the basis for the efficient functioning of their metabolism. Also, it seems to be especially important, when an organism, for example, a plant, has a photosynthetic apparatus (in chloroplasts) that is an extra endogenous source of free radicals and ROS generation.

In 1995, two independent papers confirmed the presence of melatonin in higher plants [9,10]. Since then, the search for this molecule in many edible plants and medicinal herbs has been extensively reported [28,29,30,31,32,33,34]. Melatonin has been found inter alia in apple, barley, bean, cucumber, grapes, lupine, maize, potato, rice, and tomato, among others (Table 1) [1,28,29,30,31,35,36,37,38,39].

Studies in 108 herbs species, commonly used in Chinese Medicine, revealed the presence of melatonin at concentrations ranging from a few to several thousand nanograms per gram of tissue [43], meaning that they are good natural sources of this health-related molecule. Melatonin was also detected and quantified in roots, shoots, leaves, flowers, fruits, and seeds, but its highest levels have been found in reproductive organs, particularly in seeds. This abundance is probably related to the impact of reproductive organs in plant species survival, and to their need to effectively defend them against various environmental stresses, including secondary oxidative stress [1,24]. On the other hand, it was also stated that melatonin concentrations differ not only among species, but also among different varieties of the same species [31]. Thus, it has been suggested that variations in melatonin contents might result not only from the distinct extraction and detection techniques applied, but also from the fact that both the biosynthesis and metabolism of this indoleamine are affected and modified by distinct environmental conditions [51]. This aspect partially explains why different organs of the same plant contain various amounts of melatonin during consecutive morphological and physiological stages of development. Generally, it was noticed that various plant species rich in melatonin has a greater capacity of stress tolerance [53,54,55,56], an aspect that turned attention to melatonin as a potentially effective agent in improving plant stress defense. Moreover, exogenous melatonin can also be used as a crop biostimulating agent [57,58,59].

In scientific literature, the term phytomelatonin sometimes appears, which is mostly applied to distinguish the endogenous melatonin synthesized by plants from the exogenous one, applied to them, although chemically, the molecule is the same. Not least interesting to highlight is that, in medicinal plants, including pyrethrum maruna (*Tanacetum parthenium* L.) and St. John’s wort (*Hypericum perforatum* L.), the levels of melatonin are sometimes higher than those found in animals [50], although both organisms possess the required enzymatic systems for melatonin biosynthesis (Figure 3). Moreover, and contrarily to animals, plants are also able to synthesize the melatonin precursor, tryptophan, an aromatic amino acid. Theoretically, it is constantly available for further transformations in plants, while in animals, it can only be supplied by foods. Still, and besides its in vivo synthesis, plants can also absorb melatonin provided exogenously from the environment, accumulating it at high concentrations [24,60]; thus, they can be conceived as great sources of melatonin and other indoleamines for animals.

Regarding melatonin biosynthetic pathways, they are similar between animals (humans) and plants (Figure 3). Briefly, they consist of the same precursor, tryptophan, and two common derivatives, serotonin and N-acetyl-serotonin. However, the tryptophan conversion to serotonin in animals occurs via 5-hydroxy-tryptophan, whereas in plants, it occurs via tryptamine. In the latter, serotonin can be converted into melatonin via N-acetyl-serotonin (as in animals) and 5-methoxy-tryptamine [62,63,64]. Thus, it is feasible to suppose that indole derivative transformations in plants are richer and multidirectional when compared with animals.

In all organisms, the primary function of melatonin is to act as a free radical’s scavenger and ROS detoxifying agent. These reactive species are highly generated during the aerobic metabolism, mainly in mitochondria and chloroplasts; thus, it has been hypothesized that these organelles are sites of melatonin biosynthesis in plants [45]. Indeed, it is believed that archaic purple non-sulphur bacteria (i.e., *Rhodospirillum rubrum*) and photosynthetic cyanobacteria after ingestion by primer eukaryotes eventually transformed via endosymbiosis into mitochondria and chloroplasts, respectively, or more advanced eukaryotes. According to the hypothesis of Tan et al. [45], in most organisms, if not all, the capacity of melatonin biosynthesis may have been transferred from mitochondria and chloroplasts to other cell compartments, such as endoplasmic reticulum and cytosol. In fact, during evolution, involved genes in melatonin biosynthesis found in *R. rubrum* or other non-sulphur purple bacteria and cyanobacteria were gradually integrated into the nuclear genome, although some of them may have also been lost [65]. Anyway, it seems that the hypothesis related to the original location of melatonin biosynthesis in mitochondria and chloroplasts may help in answering the question of why the melatonin content in green plants is markedly higher than in animals. Moreover, the contributory factor in explaining this phenomenon seems to be related to the presence of two organelles (mitochondria and chloroplasts) able to synthesize melatonin in plants in comparison to animals, which have only mitochondria. 

Taking a look at the predicted protective effect of melatonin on plant photosynthesis, it has also been confirmed [66,67]. Broadly, melatonin affects photosynthetic processes efficiency by the following: (1) delaying chlorophylls degradation [68,69]; (2) increasing CO_2_ uptake [70,71]; and (3) accelerating the electron transport [72,73]. However, besides its direct and indirect properties against oxidative damage, which is secondary to all abiotic and biotic stresses, melatonin also stimulates a number of specific plant defence mechanisms at the proteomic [74,75] and genetic [55,76,77] levels, against various stresses, like cold, drought, high temperatures, strong light, heavy metals, and pathogens, among others.

As a structural analog of indole-3-acetic acid (IAA), a common auxin also derived from tryptophan, melatonin, is involved in plant growth regulation and development [33,34,78]. There are also earlier reports that indicate the role of melatonin in plant circadian rhythm regulation and photoperiodic responses; however, to precisely explain these phenomena requires further investigations [23]. The studies on melatonin in plants indicate that it is a very important molecule for them. Although it is not classified as a phytohormone, melatonin does perform crucial signaling functions at a multilevel metabolic network, especially in plant–environment interactions [79]. Thus, as melatonin was identified in a huge number of edible plants and herbs, its presence in food products and beverages derived from plants is not surprising. Various beverages, for example, beer, coffee, red vine, and several types of tea [31], as well as olive and mustard oil [80], have been studied as potential natural sources of this indoleamine.

## 3. Melatonin in Humans: A Key Emphasis in Biological Activity

Circadian clocks have developed to adapt biological functions to specific times within the day or night [81]. This clock controls diurnal sleeping and waking cycles, body temperature, and hormone release. Suprachiasmatic nucleus (SCN) cells receive neural cues from the retina and send the information obtained on photoperiodic status to the pineal gland, which then synthesizes them and releases melatonin, distributing the time signal to the rest of the body [82].

With concerns to melatonin biosynthesis in humans, the exogenous amino acid tryptophan through the action of tryptophan hydroxylase (TP5H) and aromatic acid decarboxylase (AADC) enzymes is converted to the neurotransmitter, serotonin. In the subsequent step, serotonin is converted into melatonin through the influence of arylalkylamine N-acetyltransferase (AANAT) and hydroxyindole-*O*-methyltransferase (HIOMT) enzymes (Figure 3) [83].

Melatonin synthesis proceeds for 24 h/day. However, more is produced and released into the blood at night. In an adult human, approximately 30 μg of melatonin is synthesized per day, and the maximum concentration in the blood is reached at the mid-dark period. There is no storage of melatonin in the pineal gland; it is released into the bloodstream and then rapidly degraded in the liver [84]. The liver hydroxylates melatonin in the C6 position under cytochrome P450 monooxygenases A2 and 1A action, which is then converted to the sulfate derivative, a 6-sulfatoxymelatonin, which is removed from the body through urine [85].

In the circulatory system, melatonin may be bound to albumin and hemoglobin [84], but it is primarily transported by the serum albumin. The amphiphilic nature of melatonin allows it to easily cross cellular and morphophysiological barriers, including the blood–brain barrier [86]. It was shown, using 2-dimensional X-ray diffraction measurements, that melatonin can reorganize the lipid membranes, with the final effects depending on its concentration. Thus, at low concentrations, the presence of melatonin-enriched patches was observed in the cell membrane, while at high concentrations, a highly ordered uniform melatonin structure throughout the membrane became apparent. In fact, understanding this phenomenon could help explain the molecular basis of melatonin’s actions, such as its anti-amyloidogenic in the brains of Alzheimer’s disease (AD) patients [87], antioxidant and photoprotective actions, as well as the fact that melatonin so easily penetrates into different cell compartments [88]. Moreover, in mammalian cell transport systems, GLUT/SLC2A and PEPT1/2 were shown to have an active role in facilitating melatonin transport across membranes, especially into mitochondria [14].

Some studies report that foodstuffs intake containing melatonin may contribute to raising the level of this molecule in the serum and 6-sulfatoxymelatonin concentrations in urine [89]. Melatonin produced by many organs, including pineal gland, retina, gastrointestinal tract, skin, lymphocytes, and bone marrow [90], and possibly in every organ, it is also effectively taken from foodstuffs rich in this molecule [13].

As an endogenous indoleamine, melatonin has huge physiological functions, including regulation of the sleep promotion, circadian rhythms, mood, immunomodulatory actions, neuroprotective effects, bone growth, hormonal regulation, tumor suppression, defense against oxidative stress, and anti-inflammatory activity [91]. It may also be considered a therapeutic alternative to fighting bacterial, viral, and parasitic infections [91]. Generally, melatonin is not toxic and is safe; even in extreme doses, only mild adverse effects in a few individuals, such as dizziness, headache, nausea, and sleepiness, have been reported [92]. Although, because of a lack of studies, the intake of exogenous melatonin in pregnant and breastfeeding is not recommended. The main roles and functions of melatonin are presented in Figure 4. Not least important to highlight is that melatonin production in humans decreases with age (i.a., it is already depressed in women during menopause) and it seems to be especially depressed in certain diseases, such as in AD, cardiovascular disorders, and certain malignancies. Also, a reduced melatonin output has been linked to insomnia in older patients and a higher prevalence of cancer [93].

### 3.1. Regulation of the Circadian Rhythm, Biological Clock, and Sleep/Wake Cycle

One of the earliest known functions of melatonin is its impact on circadian rhythms’ regulation. Light is the environmental factor known to affect melatonin secretion. Melatonin acts directly on the SCN and modulates clock function. Circadian signals from the SCN are transmitted to the pineal gland via a multi-synaptic pathway, which involves the neural projections from the SCN to the paraventricular nuclei (PVN), the intermediolateral cell column of spinal cord (IML), superior cervical ganglion (SCG), and finally to the pineal gland (SCN → PVN → IML → SCG → pineal gland) [94]. The pineal gland is located in the midline of the brain. It transduces light and dark information to the entire body when it releases melatonin [95]. The behavioral effect of this information is a result of the specific evolutionary habits of the species. For instance, in humans, the night is a time of sleep, where melatonin promotes it and also depresses body temperature at night [96], while in many other species (e.g., wolf or owl), the nocturnal peak of melatonin secretion is associated with wakefulness and high activity.

### 3.2. Melatonin Receptors

Melatonin has specific receptors and intracellular targets in different cell types in animals to regulate many physiological functions, through modification of adenylate cyclase, guanylate cyclase, and phospholipase C activities, and membrane channels for calcium and potassium mediate melatonin signaling [97]. Specifically, the MT1 (high affinity) and MT2 (low affinity) melatonin receptors are G-protein-linked [98], and affect protein kinase activity through inhibition of adenylyl (cAMP) and guanylyl (cGMP) cyclase, respectively. These receptors regulate ion flux inside the cell via activation phospholipase [97]. Melatonin receptor 3 (MT3) is probably a quinone reductase 2 (QR2). This enzyme has an important role in free radicals’ neutralization in the organism [99]. However, MT3 receptors have not yet been found in humans, although they are expressed in various tissues of hamsters and rabbits [100]. Also, a putative nuclear receptor is the retinoid Z receptor α (RORa/RZR), though its function is still unclear [101].

Melatonin receptors are widely distributed in the body. They exist in cardiovascular, immune and endocrine systems, reproductive and gestational tissues, and even in the skin and gastrointestinal tract [100]. GPR50, a melatonin-related receptor, has also been identified and is detected in various brain structures and peripheral tissues in humans [102,103].

### 3.3. Receptor-Mediated Activities

Many of melatonin’s actions are mediated through an interaction with the G-protein coupled membrane-bound melatonin receptors MT1 and MT2, the quinone reductase II enzyme (MT3), or indirectly via nuclear orphan receptors of the RORa/RZR family [100].

Previous studies have shown that melatonin improves induced pluripotent stem cells’ (iPSCs’) proliferation and differentiation into neurons through activating phosphatidylinositide 3 kinases (PI3K)/AKT signal pathway, through modulation of MT1 and MT2 membrane receptor [104]. Moreover, melatonin is effective in treating periodontitis, mucositis, and even some cancers. Melatonin has promising antineoplastic potential owing to its antiproliferative, cytostatic, antimetastatic, and proapoptotic effects against tumor cells [105]. It enhances osseointegration and bone regeneration [76]. The ubiquitin-proteasome system controls osteoblasts and osteoclasts proliferation by providing a mechanism for regulating proteins recirculation. Also, melatonin interacts with the ubiquitin-proteasome system, thus contributing to bone regeneration regulation [106].

Melatonin, an ubiquitously-acting molecule, also regulates hormones’ release, especially those involved on seasonal reproduction and preservation of gamete quality. It inhibits the release of prolactin, luteinizing hormone, and stimulating follicle hormone during seasonal reproductive quiescence. A large body of experimental evidence indicates that melatonin has beneficial effects on reproduction in both male and female animals. In male reproduction, it regulates the following: (1) secretion of two key neurohormones, gonadotropin-releasing hormone (GnRH) and luteinizing hormone (LH), through binding to specific receptors and inhibition of both GnRH and LH production; (2) synthesis of testosterone and testicular maturation; and (3) prevention of testicular damage induced by environmental toxins or inflammation [107]. Also, experimental data demonstrated that melatonin increases the maternal-to-zygotic transition (MZT)-related genes expression in vitrified-warmed mouse MII oocytes [108].

In the recent past, several studies have highlighted the location of MT1 and MT2 receptors in different regions of the brain and retina. This has led to the manipulation of these anatomic sites specific to the disease processes [109]. MT1 and MT2 receptors have been found in all layers of the neural retina and the retinal pigmented epithelium. Melatonin protects skin cell against UV damage and regulates skin pigmentation. It also has protective effects on retinal pigment epithelial cells, photoreceptors, and ganglion cells. Moreover, when melatonin interacts with calmodulin, it directly antagonizes calcium ions binding to calmodulin [102].

Melatonin is also a compound with sufficient therapeutic benefits to prevent cardiovascular diseases [85,99]. It contributes to the cardioprotective effect of lethal ischemia-reperfusion injuries [110], and this effect is in part mediated by the activation of the tumor necrosis factor alpha (TNFα) and signal transducer and activator of transcription 3 (STAT3), which both play a key role in enhancement of the pro-survival the SAFE (survivor activator factor enhancement) pathway for cardioprotection [99].

Melatonin may also be beneficial in the treatment of dopamine-related disorders. Evidence suggests that it modulates dopaminergic pathways involved in the coordination of body movement disorders in humans [111]. Also, melatonin inhibits insulin release from β-cells in the pancreas. There is also an interrelationship between melatonin and insulin in type 1 diabetic (T1D) and type 2 diabetic (T2D) rats and humans. The effects of melatonin on insulin secretion may involve melatonin receptors (MT1 and MT2), inhibiting the adenylyl cyclase (AC)/cyclic AMP system [112]. Thus, melatonin may be promising in diabetes management via the above-mentioned processes.

### 3.4. Nonreceptor-Mediated Effects

At the whole, redox status plays an essential role in many cellular processes. ROS and RNS overproduction (generate oxidative and/or nitrosative stress) affects several biomolecules, including lipids, proteins, and DNA, consequently triggering some serious diseases, such as cancer, diabetes, and neurological and cardiologic disorders [113,114,115]. Melatonin is a highly effective ROS eliminator and exhibits numerous protective actions against oxidative stress, and is even reported to be 10× more powerful than vitamin E in scavenging ROS/RNS [116]. Melatonin receptor-independent actions include excessive ROS and RNS neutralization, generated as a result of UV and ionizing radiation, heavy metal toxicity, drugs and alcohol toxicity, and so on. Melatonin’s main metabolites include cyclic 3-hydroxymelatonin and N1-acetyl-N2-formyl-5-methoxykynuramine (AFMK) [117]. Figure 5 illustrates melatonin’s antioxidant cascade reactions in mammals [118].

Melatonin protects the brain via the immune system, and as a defense against ROS/RNS damages. It has beneficial effects against mitochondrial dysfunction [119], considered a major causative agent in neurodegenerative diseases, such as AD, Huntington’s disease (HD), and Parkinson’s disease (PD) [85,120,121]. Some studies indicate that melatonin in the range of 50–100 mg/day is effective in reducing oxidative mitochondrial dysfunction in experimental models of AD, HD, and PD. It is equally effective, in vitro and in vivo, in preventing oxidative/nitrosative stress-induced mitochondrial dysfunctions [85]. Lipid peroxidation and ROS/RNS generation in neurons also lead to neuropsychiatric and neurodegenerative disorders; thus, because of the strong antioxidant activity and diverse pleiotropic actions of melatonin, it has been proposed an effective neuroprotective agent in diverse neuropsychiatric disorders [122].

Recently, melatonin has been proposed as a potent mitochondrial protector [14,123]. Indeed, melatonin was detected at higher concentrations in mitochondria than in other organelles or subcellular compartments. To highlight that, mitochondria comprise the major cellular site of ROS production during respiration and adenosine triphosphate (ATP) synthesis. In addition, high levels of melatonin are accumulated in these organelles against a concentration gradient by mitochondrial membrane transporters. Not least important to highlight is that, according to the endosymbiotic theory on the origin of mitochondria from melatonin-producing bacteria, these organelles likely evolutionarily inherited the capacity to synthesize melatonin. Therefore, as recently reported in excellent reviews, melatonin can scavenge ROS, thus decreasing mitochondrial oxidative damages, consequently preserving their structural and functional integrity, and finally delaying cellular aging and the onset of age-related disorders [13,123].

Melatonin also possesses anti-inflammatory and anti-apoptotic properties. It protects cells against damage and improves cell survival and proliferation [124]. Further, melatonin, owing to its low toxicity and high efficacy in reducing oxidative damage, has been recommended as a protective candidate against typical chemical weapons [125] and even as a snakebite therapy [126].

As an anti-inflammatory molecule, melatonin differentially modulates pro-inflammatory enzymes; controls inflammatory mediators’ production, such as cytokines and leukotrienes; and regulates leukocytes lifespan through interfering with apoptotic processes. Indeed, it selectively inhibits inflammation by stimulating pro-inflammatory mediators, such as arachidonic and 5-hydroxyeicosatetraenoic (5-HETE) acids, through phospholipase A2 (PLA2) and 5-lipoxygenase (5-LOX) activation, respectively [83].

The experimental evidence has yielded promising results regarding the role of melatonin as an anti-carcinogenic agent [105]. It is beneficial in preventing and treating several common cancers [127]. Indeed, melatonin appears to promote apoptotic death in some cancer cells, while being protective to normal cells [107]. Melatonin has cytoprotective properties in the reproductive system by reducing apoptosis of the oocyte and granulosa cells [86]. Melatonin also inhibits the hypoxia-inducible factor 1 alpha (HIF-1α) and HIF-1α-inducible gene [128].

In conclusion, melatonin has attracted the attention of many scientists owing to its unique physiological functions and pharmacological properties. In the last decade, hundreds of scientific publications have focused on melatonin. It is a regulatory molecule that orchestrates a variety of physiological and pathophysiological processes. Beneficial effects of melatonin in various models of disease provide support/robustness for the general comment that “melatonin helps against everything” [129]. Melatonin is used by millions of people around the world to potentially retard aging diseases, improve sleep, mitigate jet-lag symptoms, and treat depression [130], besides being a safe, cheap, natural, and widely available molecule with multiple therapeutic benefits.

## 4. Melatonin Supplementation and its Health Effects for Humans

Individuals living in highly industrialized countries look for melatonin-supplemented drugs and pills as a way to improve sleep and rest, as the main advantages include quicker times to fall asleep and quicker ”jet-lag” reductions. On the other hand, some of the users also referred to experience headaches, dizziness, and sleepiness during daylight hours (overdose symptoms), but these made be placebo effects [96]. Initially, many studies have shown that exogenous melatonin was able to improve sleep, probably because of the chronobiotic role of this molecule. Indeed, it aids in the adjustment and maintenance of a regular circadian rhythmicity [131]. However, there are many other potential health effects attributed to melatonin’s application. For example, melatonin supplementation is helpful in inflammatory markers, hypertension, oxidative stress, and metabolic syndrome [132]. In one of these studies, a positive correlation was found between melatonin-rich foods and clinical-metabolic indicators. Also, melatonin concentration varies markedly among various foods, and its concentration may change during the food products’ preparation [133].

### 4.1. Melatonin and Inflammation

Melatonin inhibits cyclooxygenase-2 (COX-2), the enzyme responsible for the inflammatory cascade. Moreover, melatonin decreases pain perception during the inflammatory response, but may also enhance the analgesic effects of nonsteroidal anti-inflammatory drugs (NSAIDs), such as ibuprofen [134]. In 2002, El-Shenawy and coworkers determined the mechanism of anti-inflammatory and anti-nociceptive (sensory nervous system response to pain) effects of melatonin [135]. For that, the authors injected carrageenan (1%) into the rats’ paw to induce inflammation. Rats either received a diluent or melatonin 30 min pre-injection and they analyzed its potential for paw edema at 1, 2, 3, and 4 h post-carrageenan injection. Paw thickness aided in determining the inflammation level and edema formation. Nociception was tested based on rat vocalization after electrical stimulation of the tail. Melatonin, given intraperitoneally to rats 30 min before the carrageenan injection, was found to decrease swelling of the paw induced by the toxin. At doses of 0.5 and 1 mg/kg, melatonin was found to inhibit carrageenan-induced edema in 20.5% and 29.6%, respectively, when compared with the control at 4 h post-carrageenan injection. Melatonin at both 0.5 and 1 mg/kg also exerted anti-nociceptive effects on electrical stimulation in the rat trail test, and the increase in nociceptive thresholds to pain brought on by the electrical pain at 4 h post-treatment was 29.6 and 39.6%, respectively. When melatonin was given in combination with a COX-1 and COX-2 inhibitor, indomethacin (5 mg/kg, i.p.), 30 min in advance of carrageenan injection, the anti-inflammatory effects were enhanced by 23% in the paw edema model. A higher melatonin dose (5 mg/kg) further increased the anti-nociceptive effects of indomethacin. Anti-inflammatory and anti-nociceptive effects of the COX inhibitor slightly increased with 0.5 mg/kg melatonin. Also, it was also observed that melatonin elevates the cysteamine (300 mg/kg, s.c.) effect in the carrageenan-induced rat paw edema model. No anti-inflammatory effects were found using melatonin doses of 20 and 40 µg per paw. On the basis of the results, the authors concluded that melatonin has anti-inflammatory and anti-nociceptive effects in rats and may enhance the action of indomethacin [135].

### 4.2. Melatonin and Wound Repair

Melatonin promotes the immune system response early in the wound healing process and may even aid in healing and scar formation quality. In the study of Pugazhenthi et al. [136] with male rats, a positive correlation between superficial application of melatonin and scar repair was noted. Melatonin acts as a strong anti-inflammatory agent, also having positive immunomodulatory effects under certain conditions. In the cited study, a melatonin injection (1.2 mg/kg) into subdermal tissue in rats was examined for its ability to inhibit scar formation after an incision made in the skin. Melatonin treatment improved the scars’ quality by modulating their maturity rate and orientation of the collagen fibers. Moreover, melatonin depressed nitric oxide (NO) synthesis during the inflammatory process. In addition, while NO is generally detrimental during inflammation, it has positive effects during new tissue formation [136]. The subdermal injection of melatonin also expedited the angiogenic process, through intensifying the new blood vessel formation and increasing the vascular endothelial growth factor (VEGF) protein expression in granulation tissue formation. An increase in arginase levels, the enzyme that plays a crucial role in proline biosynthesis [136], was similarly observed. This amino acid is a factor determining a protein’s quaternary structure, and is very useful during collagen synthesis and its spatial conformation. This study documents that melatonin, when added under the epidermis, might be beneficial in improving incision healing and in reducing scar formation.

### 4.3. Melatonin and Brain Injury

Brain injury, as a result of stroke and trauma, readily interferes with neurological functions. In brain injury, inflammation and oxidative stress are commonly involved. Many brain traumas result in a change in blood flow, which leads to hypoxia. In addition, hypoxia may paradoxically lead to the so-called “oxygen burst”, which means its incomplete reduction and generation of ROS and free radicals, which in turn damages cell membranes and impairs cognitive function [137,138]. Melatonin is also a highly effective antioxidant, free radical scavenger, and anti-inflammatory agent with therapeutic effects on brain injuries [139,140,141]. Treatment with melatonin over 30 days in mice with induced stroke led to a marked improvement in brain cell survival and functional recovery. Several studies have correlated melatonin administration with the long-lasting improvement of motor and coordination physiology, which are also common in human strokes [142]. Melatonin levels often drop in patients who have suffered a severe brain injury, and animal studies have concluded that replenishing melatonin levels after the injury helps to reduce tissue injuries and also decrease cognitive deficits [143]. Melatonin also aids in toxic ROS reduction, produced when brain cells are deprived of oxygen, and likewise reduces the inflammatory systems activation in the brain; these changes are essential in aiding infant size post-head injury reduction [144]. Melatonin also has a positive impact on memory deficits when immediately administered after rabbits experience a brain injury [145]. Melatonin also protects fetal animals’ brains against ischemia/repulsion injury when it occurs during pregnancy [146]. In human pregnancies, similar injuries may result in cerebral palsy or mental retardation in newborn infants. Melatonin has been found to prevent learning disorders correlated with brain injuries in infant animals, and repeated doses have exhibited a further reduction in brain injury [123]. In fact, numerous reports have confirmed that melatonin supplementation has strong abilities to protect the brain against injury.

### 4.4. Melatonin and Cardio- and Neuro-Protection

Melatonin also increases the antioxidant enzymes’ activity and also scavenges free radicals produced during cardiac injury. Because of its direct and indirect antioxidant properties, melatonin prevents oxidative damage and reduces the size area of heart cell death [147]. Moreover, melatonin stimulates antioxidant enzymes, while simultaneously down-regulating the pro-oxidant enzymes. Its administration has also been shown to be effective in reducing hypertension and drugs-induced cardio-toxicity [147,148]. Moreover, when myocardial infarction (MI) was induced in rats, melatonin administration (4.5 mg/kg/day) post-MI (via subcutaneous osmotic pumps) significantly lowered the left ventricular (LV) levels of mRNAs, dihydropyridine receptor (DHPR), ryanodine receptor 2 (RYR2), and sarco-endoplasmic reticulum Ca^2+^-ATPase (SERCA2) compared with these measures in control rats. These authors concluded that melatonin improved the strength of the heart’s pumping action post-MI under these experimental conditions [96]. Reiter and Tan [147] have also noted that melatonin protects mitochondria, which is the energy source for cardiac muscle function.

Research focused on the protective role of melatonin in the degree of cardiac injury in patients undergoing bypass surgery, in 45 patients (45–65 years old), reported that melatonin (10 mg or 20 mg capsule once daily) increased the ejection fraction associated with a significant decline in heart rate [149]. In addition, melatonin significantly reduced plasma levels of troponin-I, interleukin-1beta, inducible oxide synthase, and caspase-3 enzymes, in both supplemented groups. The authors concluded that melatonin supplementation can ameliorate the degree of myocardial ischemic-reperfusion injury. Also, a study focusing on melatonin effects in 97 normotensive and hypertensive volunteers (63–91 years old) found that melatonin supplementation (1.5 mg/day for two weeks) had a direct hypotensive effect [150]. Also, melatonin stabilized the internal temporal order, enhancing the circadian component and the synchronization between rhythms of different physiological functions.

On the other hand, melatonin administration has also been shown to be useful to improve the cognitive performance in subjects with mild cognitive impairment (MCI) [151]. Each year, about 12% of MCI patients eventually develop AD or other types of dementia. In a study of 25 MCI patients, who received 3–9 mg of melatonin before sleep over 8–9 months exhibited better performance in the Mini-Mental State Examination and a cognitive subscale of AD assessment. The authors also reported that melatonin-supplemented patients had better scores on Rey’s verbal tests, Trail A and B tasks, and Mattis’ test, but the treatment did not affect the Digit-symbol test score. Finally, it was found that abnormally high Beck Depression Inventory scores decreased in melatonin-treated patients. Thus, these studies suggest that melatonin may be useful in the treatment of MCI patients [151]. 

The use of melatonin to treat neurodegenerative diseases revealed that this molecule was able to reduce amyloid-beta peptide (Aβ) deposits in the brains of AD patients [87]. This finding indicates that activated glia due to chronic inflammation correlates with Aβ deposits in the brains of AD patients. In this study, melatonin’ effects were specifically examined on glia activation and an increase in learning in amnesic rats, induced by Aβ peptide 25–35, was observed. Cognitive function was analyzed using the Morris water maze test. The study reported that Aβ 25–35 injected into the rat hippocampus promoted impairments in learning and memory, which was associated with a rise of activated glial cells compared to controls. When melatonin was administered at 0.01, 0.1, and 1 mg/kg over 10 days, learning and memory improved in rats that had received the Aβ 25–35. Melatonin inhibited pro-inflammatory factors in these animals, and when applied to humans, it may aid in the treatment of AD patients with regards to memory and learning improvement [87].

A report by Kilic et al. [152] was conducted to identify the potential positive effects of melatonin in amyotrophic lateral sclerosis (ALS). ALS is a disease that involves oxidative stress in alpha-motor neurons of the spinal cord. The authors observed that melatonin is a possible neuroprotective compound and strong antioxidant in this disease model. Experiments were done with different test models: (1) in cultured motoneuronal cells (NSC-34); (2) in a genetic mouse model of ALS [SOD1 (G93A)-transgenic mice], and (3) in a group of 31 patients with sporadic ALS. Melatonin decreased glutamate-induced cell death in cultured motoneurons. In SOD1 (G93A)-transgenic mice, high melatonin doses administered post-poned disease advancement orally and increased survival. Also, ALS patients receiving a high dose (300 mg/day) of rectal melatonin over the course, over two years, were found to have normalized levels of protein carbonyls (an oxidative stress marker) in circulation. The authors concluded that high melatonin doses are safe for humans and it is suitable for clinical trials to gain information on its effect as an antioxidant in ALS patients [152].

## 5. Oral Bioavailability of Plant Melatonin

Exogenously administered melatonin is well absorbed following oral administration, widely distributed, and virtually completely metabolized in humans [153]. Melatonin receptors are abundant in the brain and melatonin readily penetrates the blood–brain barrier [154]. Melatonin, when consumed as a drinking fluid [155] or taken as a galenic tablet [156], is readily absorbed into circulation. Thus, it would be expected that melatonin from foods would also likely be absorbed. While this may be true, melatonin uptake from herbal remedies or products, and phytomelatonin oral bioavailability, have not been well explored.

The study of Yeleswaram et al. [157], assessing the oral bioavailability of synthetic melatonin in rats, dogs, and monkeys, showed a dose-dependent availability that differed among the species examined. The dose normalized oral bioavailability following a 10 mg/kg oral dose of melatonin represented 53.5% in rats, while it was around 100% in dogs and monkeys. However, the oral bioavailability of melatonin in dogs was found to be decreased to 16.9% following a 1 mg/kg oral dose. In vitro permeability studies with CACO-2 cells, the model suggested that exogenous melatonin is likely to be well absorbed in humans [157]. Hattori et al [10] investigated the oral melatonin bioavailability from 24 edible plants and found that the administration of a diet consisting in plant products rich in melatonin to 48 h fasted chicks significantly increased the level of circulating melatonin, that is, the daytime melatonin levels were roughly doubled. Moreover, they found that competes with melatonin binding sites in the rabbit brain [10].

Melatonin absorbed into feed pellets or into wheat grains that received chicken food also led to its high initial plasma melatonin concentration, followed by a rapid decrease after 2–3 h; however, elevated levels were still detectable up to 24 h after administration. While melatonin absorbed into cracked wheat grains subsequently washed with ethanol eliminated the initial transitory peak, it sustained the plasma levels for at least 12 h in the normal nocturnal range (750 pM), and there was no measurable melatonin increase (<60 pM) 18 h later. The authors concluded that melatonin-treated, ethanol-rinsed cracked wheat grains could be used to experimentally mimic the long-night plasma melatonin patterns [158].

Food restriction of rats and then feeding them with regular chow or melatonin-rich (3.5 ng/g FW) walnuts (*Juglans regia* L.) was followed by elevated blood melatonin concentrations in animals that had eaten walnuts (increased from 11.5 to 38.0 pg/mL), compared with rats fed the control diet. Increases in blood, melatonin were also accompanied by rises in serum antioxidant capacity [44]. However, it is unclear whether other plant components can trigger or reduce the release or uptake of melatonin into the gastrointestinal tract or organs involved in metabolism, such as liver [159].

Similar experiments with rats were performed by Aguilera et al. [160], but as a source of phytomelatonin, they used bean (*Phaseolus vulgaris* L.) sprouts, that is, the aqueous extract from bean sprouts. The authors assessed in rats the effect of bean sprout intake on plasma melatonin levels and its metabolite, 6-sulfatoxymelatonin, in urine. Also, they compared the bioavailability derived from bean sprouts versus synthetic melatonin. Blood and urine samples were obtained before and after 90 min of melatonin (phyto- or synthetic) administration via a gavage. The plasma melatonin levels increased after bean sprout ingestion (16%, *p* < 0.05). This increment correlated with the urinary 6-sulfatoxymelatonin content, the principal biomarker of plasma melatonin levels (*p* < 0.01). In contrast to the previous discussed findings by Reiter, Manchester, and Tan [44], the antioxidant capacity did not exhibit any significant change. The comparison of bioavailability between bean sprouts phytomelatonin and synthetic melatonin indicated slightly higher levels of plasma melatonin (17%) in rats fed with the synthetic melatonin solution [160].

### Melatonin Oral Bioavailability in Humans

When looking at melatonin bioavailability, a number of studies already performed report an extensive variation, even assessing its oral and/or intravenous administration. In addition, and not least interesting to highlight, is that considerable variations in melatonin aspects related with its absorption, metabolism, and elimination have also been stated between distinct individuals. These aspects clearly reflect the need for more in-depth studies on the pharmacokinetic properties of melatonin. For instance, Andersen et al. [161] conducted a crossover cohort study to determine the pharmacokinetics of 10 mg oral melatonin or 10 mg intravenous melatonin in healthy male volunteers. Orally administered melatonin was rapidly absorbed, with T_max_ being reached at 41 min; however, the *C*_max_ and the area under the curve (*AUC)* varied significantly among volunteers. With regards to oral and intravenous melatonin elimination half-lives, they were 54 min and 39 min, respectively. Thus, oral melatonin bioavailability was found to be only 3% with considerable inter-volunteer variability.

The physiological changes in human plasma melatonin have also been investigated after beer consumption. Eighteen brands of beer containing melatonin up to 170 pg/mL were analyzed. Beer administered to seven subjects consisting of four men (660 mL) and three females (330 mL) aged between 20 and 30 years resulted in an uptake of 112 ng and 56 ng of melatonin for men and women, respectively. Melatonin bioavailability from beer was directly proportional to the dose, and serum analysis by ELISA prior and 45 min post-administration confirmed that melatonin increase in human serum was related to drinking melatonin-rich beer [162]. In humans, serum melatonin concentrations increase significantly from 10 to 12 pg/mL at 1 h after a single consumption of 100 mL red wine [80]. The “French paradox” (high long-life expectancy of the French people, despite its theoretically unhealthy diet rich in fat), associated with regular consumption of red wine, which includes plant polyphenols, especially resveratrol, may also be because of the intake of melatonin, which is present in red wine.

Fruits seem to be also a good source of phytomelatonin. Served to 12 healthy male volunteers for breakfast as either juice extracted from 1 kg of orange or pineapple or two whole bananas, containing 302 ng, 150 ng, and 1.7 ng phytomelatonin, respectively, elevated the plasma melatonin levels, generating characteristic change kinetics over time [163]. A blood sample was collected from the experimental participants before breakfast (control), as well as 1, 2, and 3 h after consuming the fruit products. Serum melatonin concentration (assessed by ELISA) increased already at 1 h, reaching the highest values at the second hour after fruit consumption. The values were significantly increased for pineapples (146 vs. 48 pg/mL, *p* = 0.002), oranges (151 vs. 40 pg/mL, *p* = 0.005), and bananas (140 vs. 32 pg/mL, *p* = 0.008), respectively. Between the second and third hours of the experiment, plasma melatonin levels decreased rapidly in all subjects, reaching values lower than 50 pg/mL after the third hour. In this experiment, serum antioxidant capacity (estimated by oxygen radical absorbance capacity (ORAC) and ferric reducing antioxidant power (FRAP) tests) following fruit consumption was also significantly elevated and was strongly correlated with serum melatonin concentration for all the three fruits tested [163].

Although many herbs containing significant melatonin amounts are known, except St. John’s wort, no study reported the oral bioavailability of melatonin from herbal products. The daily treatment of 13 healthy subjects for three weeks with a hydroethanolic extract of dried flowering tops or aerial parts of St. John’s wort significantly increased the nocturnal melatonin plasma concentration in these subjects [164]. However, this study was not very informative because the melatonin concentration in the extract, and the exact melatonin amount ingested from the plant product, was unknown. Conclusions were only based on the rise in plasma melatonin levels. Therefore, it is possible to infer that the incremental rise in melatonin levels may not have been linked to phytomelatonin intake from plant, but rather occurred as a result of some plant constituents that triggered the release of endogenous melatonin from the gut.

On the other hand, some authors criticized the studies investigating the measurement of blood and urine melatonin after the intake of foods containing this molecule, suggesting that the claimed increases in circulating melatonin are not consistent with the amount of dietary melatonin ingested [165,166]. However, after the administration of a glass of melatonin-enriched red wine, the serum melatonin levels changed, peaking at 60 min post-intake, supporting the role of absorbed dietary melatonin in counteracting the physiological decline of its endogenous levels in the bloodstream [89].

## 6. Conclusions

Therein, the use of dietary melatonin may be extremely beneficial in helping to maximize the health-promoting effects of medicinal plants and healthy foods in humans, possibly acting in synergy with other bioactive phytochemicals (i.e., polyphenols) that are ingested daily. However, the current lack of knowledge on the oral bioavailability of melatonin in the humans diet clearly indicates the need for more in-depth clinical trials, namely considering the circadian and seasonal variations of endogenous melatonin and allowing to estimate the amount of melatonin ingested. In fact, as there is no difference between endogenous and exogenously acquired melatonin, it is very difficult to assess the dietary contribution in humans.

## Figures and Tables

**Figure 1 cells-08-00681-f001:**
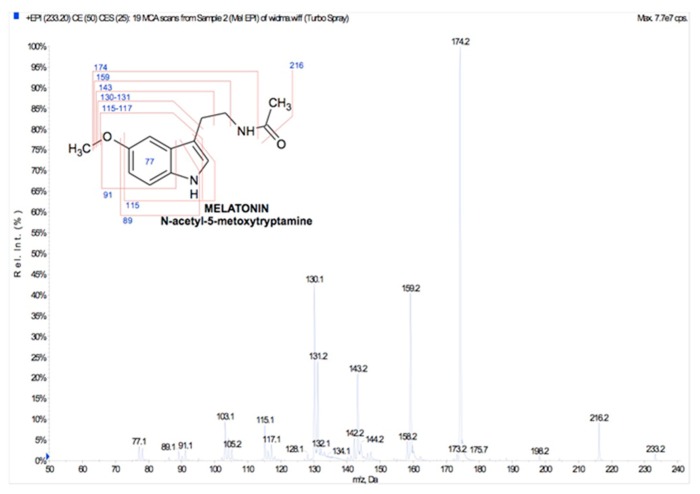
Chemical structure and mass spectrum of melatonin fragmentation pattern.

**Figure 2 cells-08-00681-f002:**
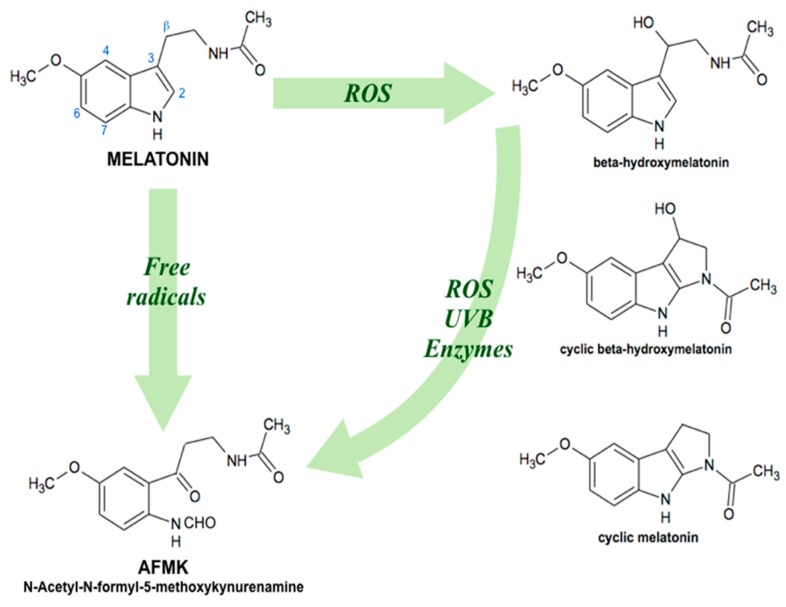
Antioxidant cascade of melatonin derivatives in plants. Melatonin can be hydroxylated at different C-atoms (2, 3, 4, 6, 7, β) by subsequent interactions with two hydroxyl radicals. ROS, reactive oxygen species; UVB, ultraviolet B (shortwave) rays.

**Figure 3 cells-08-00681-f003:**
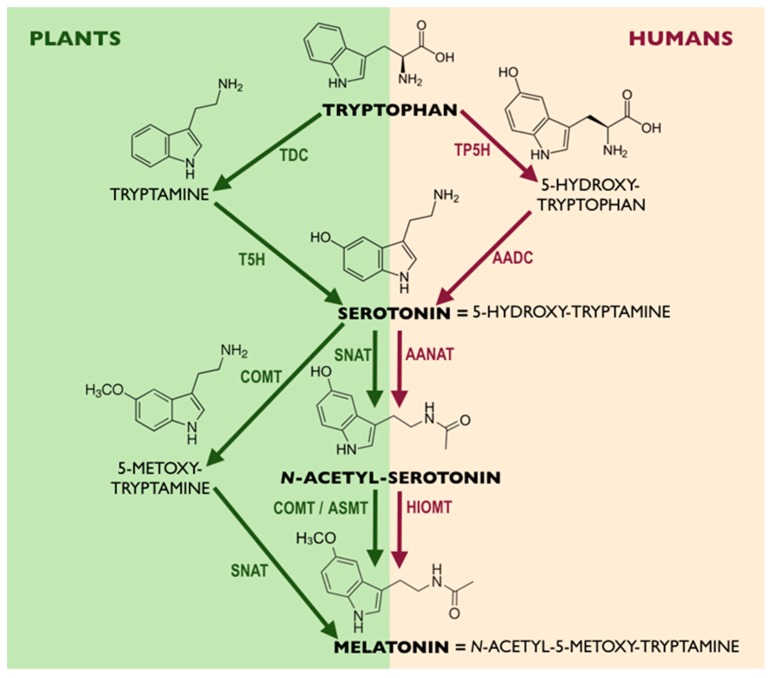
Comparison of melatonin biosynthetic pathways in plants and humans. AADC, aromatic-l-amino-acid decarboxylase; AANAT, arylalkylamine N-acetyltransferase; ASMT, N-acetylserotonin methyltransferase; COMT, caffeic acid O-methyltransferase; HIOMT, hydroxyindole-O-methyltransferase; SNAT, serotonin-N-acetyltransferase; T5H, tryptamine 5-hydroxylase; TDC, tryptophan decarboxylase; TP5H, tryptophan 5-hydroxylase [61].

**Figure 4 cells-08-00681-f004:**
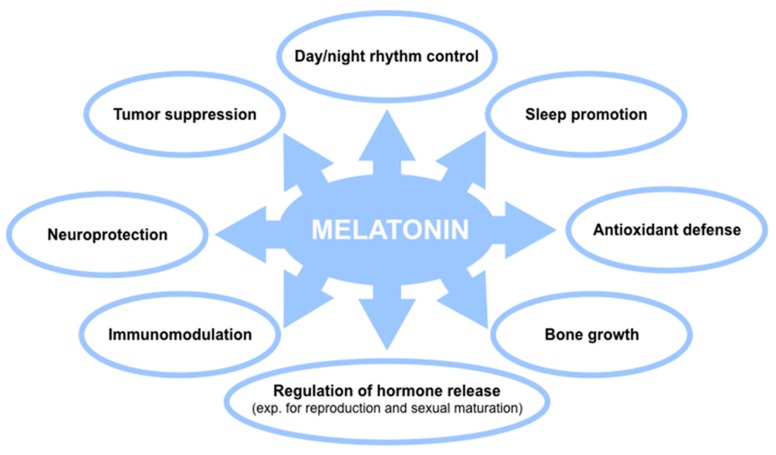
The main roles and functions of melatonin in humans.

**Figure 5 cells-08-00681-f005:**
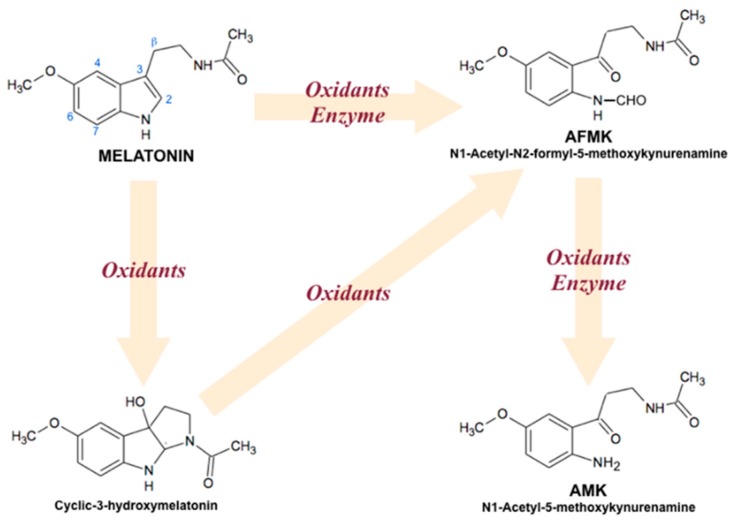
Interaction of melatonin with oxidizing agents in humans.

**Table 1 cells-08-00681-t001:** Melatonin contents in some plant organs [40].

Common Name	Latin Name	Organ	Melatonin [ng g^-1^ DW](or FW*)	Reference
Coffee robusta	*Coffea canephora Pierr.*	Bean	5800	[41]
Coffee arabica	*Coffea arabica* (L.)	Bean	6800	[41]
Black pepper	*Piper nigrum* (L.)	Leaf	1093	[42]
Wolf berry (goji)	*Lycium barbarum* (L.)	Fruit	530	[43]
White radish	*Raphanus sativus* (L.)	Bulb	485	[43]
White mustard	*Sinapis alba* (L.)	Seed	189	[1]
Black mustard	*Brassica nigra* (L.)	Seed	129	[1]
Curcuma	*Curcuma aeruginosa Roxb.*	Root	120	[43]
Wolf berry	*Lycium barbarum*	Seed	103	[1]
Burmese grape	*Baccaurea ramiflora* Lour.	Leaf	43.2	[42]
Fenugreek	*Trigonella foenum-graecum* (L.)	Seed	43	[1]
Almond	*Prunus amygdalus* (Batsch)	Seed	39	[1]
Sunflower	*Helianthus annuus* (L.)	Seed	29	[1]
Fennel	*Foeniculum vulgare* (Gilib.)	Seed	28	[1]
Agati	*Sesbania glandiflora* (L.) Desv.	Leaf	26.3	[42]
Bitter melon	*Momordica charantia* (L.)	Leaf	21.4	[42]
Alfalfa	*Medicago sativum* (L.)	Seed	16	[1]
Green cardamom	*Elettaria cardamomum* (White et Maton)	Seed	15	[1]
Flax	*Linum usitatissimum* (L.)	Seed	12	[1]
Linseed (flax)	*Linum usitatissimum* (L.)	Seed	12	[1]
Java bean	*Senna tora* (L.) Roxb.	Leaf	10.5	[42]
Sesban	*Sesbania sesban* (L.) Merr.	Leaf	8.7	[42]
Anise	*Pimpinela anisum (L.)*	Seed	7	[1]
Celery	*Apium graveolens* (L.)	Seed	7	[1]
Coriander	*Coriandrum sativum* (L.)	Seed	7	[1]
Poppy	*Papaver somniferum* (L.)	Seed	6	[1]
Walnut	*Juglans regia* (L.)	Seed	3.5	[44]
Milk thistle	*Silybum marianum* (L.)	Seed	2	[1]
Sweet cherries	*Prunus avium* (L.)	Fruit	120*	[45]
Tart cherries	*Prunus cerasus* (L.)	Fruit	19.5*	[46]
Grapevine	*Vitis vinifera* (L.)	Fruit	18*	[47]
Cherry	*Prunus cerasus* (L.)	Fruit	18*	[46]
Corn	*Zea mays* (L)	Seed	14-53*	[24]
Cucumber	Cucumis sativus (L)	Seed	11-80*	[24]
Strawberry	*Fragaria x ananassa* (Duch.)	Fruit	11.3*	[48]
Pomegranate	*Punica granatum* (L.)	Fruit	5.5*	[49]
Tall fescue	*Festuca arundinacea*	Seed	5.3*	[10]
St. John’s wort	*Hypericum perforatum* (L.)	Flower	4*	[50]
Lupine	*Lupinus albus* (L.)	Seed	3.8*	[36]
Tomato	*Solanum lycopersicum* (L.)	Fruit	2.5*	[51]
Fever few	*Tanacetum parthenium* (L.)	Leaf	2*	[50]
St. John’s wort	*Hypericum perforatum* (L.)	Leaf	2*	[50]
Oat	*Avena sativa* (L.)	Seed	1.8*	[10]
Corn	*Zea mays* (L.)	Seed	1.4*	[10]
Grapevine	*Vitis vinifera* (L.)	Fruit	1.2*	[52]
Rice	*Oryza sativa japonica* (L.)	Seed	1*	[10]

* corresponds to FW. DW, dry weight; FW, fresh weight.

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
