# Peer review of "Melatonin in Medicinal and Food Plants: Occurrence, Bioavailability, and Health Potential for Humans"

_cells, 2019, doi:10.3390/cells8070681_

Round 1
Reviewer 1 Report
This is an extensive, albeit largely inaccurate, review of the literature concerning plant melatonin and its effects.It is full of fundamental mistakes that do not derive from English language problems. For example, in the second sentence, the authors claim melatonin has been identified in a wide array of organisms. They list "Bacteria, Euglenoidea, Alveolata, Ochrophyta. Rhodophyta, Viridoplantae, Fungi and Metazoa". First, this isn't true, but secondly, the statement is biologically naive. Bacteria is no longer considered a taxonomic group, but lets give the authors that it is a Domain. Eugleoidea is a Class in the Superphylum Exocata, Alveolata is a Superphylum. Ochrophyta and Rhodophyta are Phyla. Viridoplantae is not considered a taxonomic terms for green plants. Fungi and Animalia are Kingdoms. The description of biological rhythmicity is completely incorrect. First, on line 174, the authors state that archaea have a circadian clock. They might, but it has not been demonstrated. The citation they use is Alberts et al Molecular Biology textbook, which is not appropriate. Secondly they state that "the circadian clock depends on the cyclical change of day and night". That's actually the opposite of true. The circadian clock is DEFINED by its capacity to oscillate in the absence of light dark cycles. It is then entrained to local time via the retinal pathways in mammals and extraretinal pathways in other species. Again, the authors cite the Alberts textbook. This, again is wholly inappropriate. However, inappropriate citation is the most consistent feature of this paper. For the idea that plants synthesize melatonin, the authors cite a book by Ravishankar and Ramakrishna, which is not about biosynthesis, and two papers about the effects of melatonin, not biosynthesis. For the idea that melatonin is produced in many organs, the authors cite a Pandi-Perumal paper about the broad effects of melatonin and the distribution of receptors; again, not biosynthesis. I have no doubt Pandi-Perumal has published work about widespread biosynthesis. I know him. It just isn't this paper. In fact, in nearly every case I check for citation, it is the incorrect citation.
Author Response
This is an extensive, albeit largely inaccurate, review of the literature concerning plant melatonin and its effects. It is full of fundamental mistakes that do not derive from English language problems. For example, in the second sentence, the authors claim melatonin has been identified in a wide array of organisms. They list "Bacteria, Euglenoidea, Alveolata, Ochrophyta. Rhodophyta, Viridoplantae, Fungi and Metazoa". First, this isn't true, but secondly, the statement is biologically naive. Bacteria is no longer considered a taxonomic group, but let’s give the authors that it is a Domain. Eugleoidea is a Class in the Superphylum Exocata, Alveolata is a Superphylum. Ochrophyta and Rhodophyta are Phyla. Viridoplantae is not considered a taxonomic terms for green plants. Fungi and Animalia are Kingdoms.
Reply: Thanks for your observation, we deleted this erroneous statement.
The description of biological rhythmicity is completely incorrect. First, on line 174, the authors state that archaea have a circadian clock. They might, but it has not been demonstrated. The citation they use is Alberts et al Molecular Biology textbook, which is not appropriate. Secondly they state that "the circadian clock depends on the cyclical change of day and night". That's actually the opposite of true. The circadian clock is DEFINED by its capacity to oscillate in the absence of light dark cycles. It is then entrained to local time via the retinal pathways in mammals and extraretinal pathways in other species. Again, the authors cite the Alberts textbook. This, again is wholly inappropriate.
Reply: We changed these sentences and inserted a new reference.
Pfeffer, M.; Korf, H.W.; Wicht, H. Synchronizing effects of melatonin on diurnal and circadian rhythms. General and comparative endocrinology 2018, 258, 215-221.
However, inappropriate citation is the most consistent feature of this paper. For the idea that plants synthesize melatonin, the authors cite a book by Ravishankar and Ramakrishna, which is not about biosynthesis, and two papers about the effects of melatonin, not biosynthesis.
Reply: We changed these reference for a new one:
Zhao, D.; Yu, Y.; Shen, Y.; Liu, Q.; Zhao, Z.; Sharma, R.; Reiter, R.J. Melatonin synthesis and function: Evolutionary history in animals and plants. Front Endocrinol (Lausanne) 2019, 10, 249-249.
For the idea that melatonin is produced in many organs, the authors cite a Pandi-Perumal paper about the broad effects of melatonin and the distribution of receptors; again, not biosynthesis. I have no doubt Pandi-Perumal has published work about widespread biosynthesis. I know him. It just isn't this paper. In fact, in nearly every case I check for citation, it is the incorrect citation.
Reply: We changed this reference for other Pandi-Perumal paper.
Pandi-Perumal, S.R.; Srinivasan, V.; Maestroni, G.J.; Cardinali, D.P.; Poeggeler, B.; Hardeland, R. Melatonin: Nature's most versatile biological signal? The FEBS journal 2006, 273, 2813-2838.
This work says “Synthesis of melatonin also occurs in other areas of the body, including the retina, the gastrointestinal tract, skin, bone marrow and in lymphocytes, from which it may influence other physiological functions through paracrine signaling.”
Reviewer 2 Report
Manuscript ID: cells-535965 Title: Melatonin in Medicinal and Food Plants: Occurrence, Bioavailability and Health Potential for Humans Authors clearly described the melatonin contents in some plants and demonstrated the synthesis, degradation, and biological activity. This review helps to reader to understand the overall role of melatonin in the biological system. I have only few comments about this manuscript. First of all, authors should describe the bioavailability of exogenous melatonin in the animal and humans based on ADME (absorption, distribution, metabolism, and excretion) and penetration ratio of blood-brain barrier. In addition, it is helpful to demonstrate the schematic drawing of melatonin receptor in the body and brain regions.Author Response
Manuscript ID: cells-535965 Title: Melatonin in Medicinal and Food Plants: Occurrence, Bioavailability and Health Potential for Humans Authors clearly described the melatonin contents in some plants and demonstrated the synthesis, degradation, and biological activity. This review helps to reader to understand the overall role of melatonin in the biological system. I have only few comments about this manuscript. First of all, authors should describe the bioavailability of exogenous melatonin in the animal and humans based on ADME (absorption, distribution, metabolism, and excretion) and penetration ratio of blood-brain barrier. In addition, it is helpful to demonstrate the schematic drawing of melatonin receptor in the body and brain regions.
Reply: In accordance, in section 5. Oral Bioavailability of Plant Melatonin, we inserted the next sentences:
Exogenously administered melatonin is well-absorbed following oral administration, widely distributed and virtually completely metabolized in humans [154]. Melatonin receptors are abundant in the brain and melatonin readily penetrates the blood-brain barrier [155].
154. Therapeutic Goods Administration. Australian Public Assessment Report for Melatonin. AusPAR Circadin Melatonin Commercial Eyes Pty Ltd. Department of Health and Agein. Australian Government. 2009.
155. Ng, K.Y.; Leong, M.K.; Liang, H.; Paxinos, G. Melatonin receptors: distribution in mammalian brain and their respective putative functions. Brain structure & function 2017, 222, 2921-2939, doi:10.1007/s00429-017-1439-6.
Also, about melatonin receptors the readers can see previous sections as:
3.2. Melatonin Receptors
3.3. Receptor-Mediated Activities
3.4. Nonreceptor-Mediated Effects
Reviewer 3 Report
This review is showing function of melatonin which is a widespread molecule among living organisms and contributes to multiple biological, hormonal and physiological processes. This review is well written and the information described is updated.
Author Response
This review is showing function of melatonin which is a widespread molecule among living organisms and contributes to multiple biological, hormonal and physiological processes. This review is well written and the information described is updated.
Reply: Thanks for appreciate our work.